# Green synthesis of olefin-linked covalent organic frameworks for hydrogen fuel cell applications

Zhifang Wang[1,2,6], Yi Yang[1,2,6], Zhengfeng Zhao[1,3,6], Penghui Zhang[1], Yushu Zhang[1], Jinjin Liu[1], Shengqian Ma [4], Peng Cheng[1,2,5], Yao Chen[1,3] & Zhenjie Zhang [1,2,5✉]

Green synthesis of crystalline porous materials for energy-related applications is of great significance but very challenging. Here, we create a green strategy to fabricate a highly crystalline olefin-linked pyrazine-based covalent organic framework (COF) with high robustness and porosity under solvent-free conditions. The abundant nitrogen sites, high hydrophilicity, and well-defined one-dimensional nanochannels make the resulting COF an ideal platform to confine and stabilize the $H_3PO_4$ network in the pores through hydrogen-bonding interactions. The resulting material exhibits low activation energy ($E_a$) of 0.06 eV, and ultrahigh proton conductivity across a wide relative humidity (10–90 %) and temperature range (25–80 °C). A realistic proton exchange membrane fuel cell using the olefin-linked COF as the solid electrolyte achieve a maximum power of 135 mW cm$^{-2}$ and a current density of 676 mA cm$^{-2}$, which exceeds all reported COF materials.

[1] State Key Laboratory of Medicinal Chemical Biology, College of Chemistry, Nankai University, Tianjin, China. [2] Key Laboratory of Advanced Energy Materials Chemistry, Ministry of Education, Nankai University, Tianjin, China. [3] College of Pharmacy, Nankai University, Tianjin, China. [4] Department of Chemistry, University of North Texas, Denton, TX, USA. [5] Renewable energy conversion and storage center, Nankai University, Tianjin, China. [6]These authors contributed equally: Zhifang Wang, Yi Yang, Zhengfeng Zhao. ✉email: zhangzhenjie@nankai.edu.cn

The growing concerns of fossil fuel exhaustion and global environmental problems have prompted the exigent pursuit of alternative energy resources. The proton-exchange membrane fuel cells (PEMFCs) have been considered as an ideal candidate for sustainable energy with clean combustion products[1]. Proton exchange membranes (PEMs) are the core components in fuel cells, and their proton conductivity is critical to the performance of fuel cells. Thus, the development of membrane materials with high proton conductivity is a campaign in this field. In the past few decades, rational design and synthesis of organic polymers as proton-conducting materials have aroused great concern due to their good structure robustness and facile membrane fabrication[2]. Although the first report of Nafion (sulfonated tetrafluoroethylene-based fluoropolymer-copolymer) can be tracked back to as early as the late 1960s, the standard membranes for PEMFC are still dominated by Nafion and its derivatives[3,4]. However, some unsolved challenges such as high cost, high water/methanol permeability, and narrow working conditions of Nafion membranes hinder the commercialization of fuel cells[5]. Moreover, their low-ordered structures prevent precise structural design and understanding of the relationship between structures and properties[6]. Thus, developing alternative polymeric systems for PEMFC is in urgent demand.

Covalent organic frameworks (COFs) are an emerging class of porous crystalline organic polymers built from the covalent linkage of geometrically predefined organic building blocks into extended two or three-dimensional (2D or 3D) networks[7–10]. Attributed to their well-defined structures, low densities, high surface areas, and facile functionality, COFs have been considered as highly designable and functionalized materials platforms with superior potential in diverse applications, such as gas adsorption and separation, catalysis, molecular sensing, and optoelectronics[11–14]. Notably, the uniform and tunable channels of 2D COFs are very helpful to enhance the proton mobility and the proton carrier loading capacity, which have been explored for proton-conducting applications[15–21]. The incorporation of small acidic molecules, including phytic acid[22], phosphoric acid[23–27], p-toluene sulfonic acid[28], etc., into the channels of 2D COFs has been proved to one of the successful strategies to fabricate proton-conducting materials. However, most of the current COFs are constructed by reversible covalent bonds, such as boroxine, boronate ester, imine, and hydrazone, which usually exhibit inherent low stability especially in the acidic condition that seriously limits their practical applications toward fuel cells under harsh operating conditions.

Recently, olefin-linked 2D COFs synthesized via Knoevenagel condensation or Aldol condensation reaction have attracted continuous attention (Supplementary Table 1)[29–32]. The irreversibility of the C=C double bond linkages guarantee high chemical stability for olefin-linked COFs, but make it challenging to obtain highly crystalline materials. At present, all olefin-linked COFs were fabricated via the solvothermal synthesis in the presence of suitable organic solvents and special catalysts (Fig. 1). For instance, the Jiang group synthesized cyanovinylene linkage COFs in a mixture of mesitylene and dioxane with NaOH as a catalyst[30]. Zhang et al. reported a series of olefin-linked pyridine-based COFs in DMF solution with piperidine as a catalyst[33]. Yaghi[34] and other groups[35] successfully synthesized olefin-linked triazine-based COFs in a mixture of mesitylene, 1,4-dioxane, and acetonitrile, with trifluoroacetic acid as a catalyst, which exhibited high proton conductivity after doping H₂SO₄. All the reported synthesis required suitable organic solvent as synthesis mediums, which is often a time-consuming process and environmentally unfriendly. Meanwhile, the traditional solvothermal reaction was usually carried out at high temperature and pressure in a sealed pyrex tube, which hindered their scalable production. To overcome these challenges, solvent-free mechanohemical[36] and ambient temperature ionothermal[37] as simple and green synthetic routes[38] have been developed to fabricate β-ketoenamine-linked or imine-linked COFs. Undoubtedly, developing facile and green synthetic routes for fabricating robust olefin-linked COFs is highly desirable, especially for industrial-scale production and applications.

Herein, we reported a green strategy to construct a olefin-linked NKCOF-10 (NKCOF = Nankai covalent organic framework) through benzoic-anhydride-catalyzed Aldol reaction between the activated methyl groups of 2,5-dimethylpyrazine (PZ) and 1,3,5-triformylbenzene (TFB) under solvent-free conditions. The resulting olefin-linked COF was of a layered honeycomb-like crystalline framework with a high surface area and exhibited exceptional stability toward harsh conditions, including strong acid/base. Moreover, we anchored the proton carriers, H₃PO₄, within the pores of NKCOF-10 through pyrazine functionalities to fabricate H₃PO₄@NKCOF-10, which exhibited ultrahigh proton conductivity and excellent performance as solid electrolyte membranes under real fuel cell-operating conditions.

## Results

**Design and synthesis of olefin-linked pyrazine-based COFs.** In order to explore the green synthesis conditions for olefin-linked COFs, a model compound (2,5-distyrylpyrazine) was firstly synthesized via the condensation reaction of PZ with benzaldehyde catalyzed by benzoic anhydride[39]. The successful synthesis of the model compound was proved by ¹H NMR and Fourier transform infrared (FT-IR) spectroscopy (Supplementary Figs. 1–3). Inspired by this reaction, the 2-connected linear monomer of PZ and 3-connected monomer of TFB with $C_3$ symmetry was judiciously chosen to synthesize the target olefin-linked COF with hcb topology (Fig. 2a). A solvent-free reaction of PZ and TFB with benzoic anhydride as a catalyst under 200 °C for 5 days afforded a green-yellow monolith, which has not been observed in the COF field yet (Supplementary Fig. 4). This exciting result inspired us to use reaction containers with different shapes to control the shapes of the formed monoblock (e.g., cylinder, pellet). This discovery demonstrates great potential to fabricate continuous COF monoliths toward special applications. The formed samples were thoroughly washed with methanol to remove the benzoic anhydride and then dried at 80 °C to obtain the purified NKCOF-10. The crystalline structure of NKCOF-10 was examined by powder X-ray diffraction (PXRD) patterns. As shown in Fig. 2b and Supplementary Fig. 5, the existence of a narrow fwhm₁₀₀ of 0.81° in the PXRD pattern indicated that NKCOF-10 had a high degree of crystallinity with long-range ordering in the frameworks. Pawley refinements was performed with the data from the PXRD pattern to yield unit cell parameters of $a = b = 23.06$ Å, $c = 3.46$ Å, $\alpha = \beta = 90°$, $\gamma = 120°$ with $R_{wp} = 3.58\%$ and $R_p = 2.69\%$. Peaks at 4.4°, 7.65°, 9.02°, and 25.7° correspond to the (100), (110), (200), and (001) planes, respectively, which were in good agreement with the simulated AA-stacking model.

The chemical structure of NKCOF-10 was confirmed through FT-IR spectroscopy, solid-state ¹³C NMR spectroscopy. In the FT-IR spectrum, NKCOF-10 showed a new characteristic absorbance at 1630 cm⁻¹, confirming the formation of the C=C bonds in the as-prepared COF skeletons (Fig. 3a). The C–H stretching vibration peak of methyl (3000 cm⁻¹) of the PZ monomer disappeared, and the C=O stretching vibration of the TFB monomer was attenuated in NKCOF-10, indicative of the polymerization. Moreover, the complete removal of benzoic anhydride via washing with methanol was verified in NKCOF-10 via FT-IR (Supplementary Fig. 6). Comparing the solid-state ¹³C NMR spectrum of

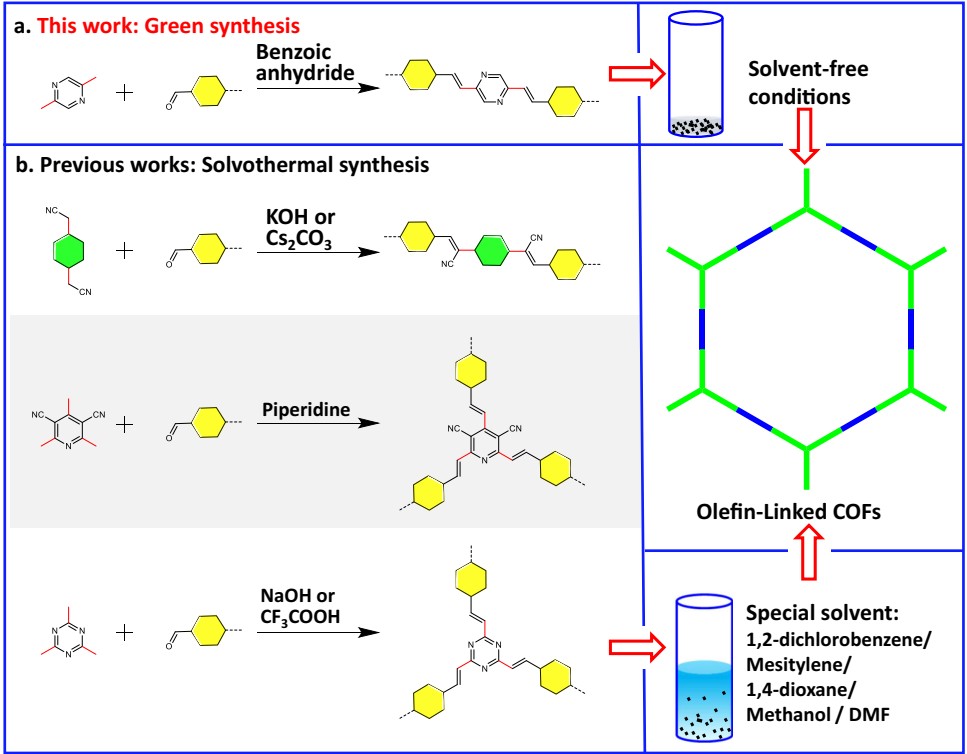

**Fig. 1 Design and reactions. a** Schematic representation of olefin-linked COFs through green synthesis. **b** Previous works of 2D olefin-linked COFs through the solvothermal synthesis.

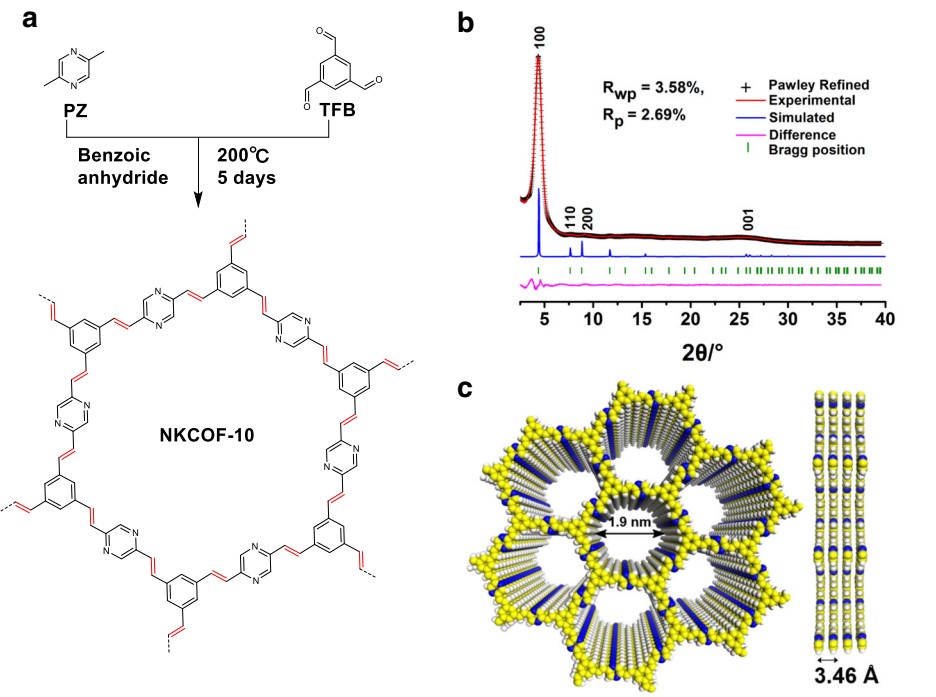

**Fig. 2 Structure and crystallinity. a** Structure and synthesis of NKCOF-10. **b** PXRD patterns of NKCOF-10: experimental patterns (red), pawley-refined profile (black), the simulated pattern for eclipsed AA stacking mode (blue), and the refinement differences (pink). **c** Top and side views of the eclipsed AA-stacking model.

NKCOF-10 with the liquid $^{13}$C NMR spectra of the model compound (2,5-distyrylpyrazine) revealed that these signals matched the corresponding carbon atoms. The two expected peaks at 125 and 136 ppm were assigned to the vinylene (C=C) carbons, which further supported the formation of olefin-linked COFs. In addition, the overlapping peaks from 126 to 133 ppm were assigned to phenyl carbons (Fig. 3b). High-resolution transmission electron microscopy (HRTEM) images of NKCOF-10 revealed an interlayer distance of ~2.9 Å (Fig. 3d and Supplementary Fig. 7), which corresponds to the (001) facet.

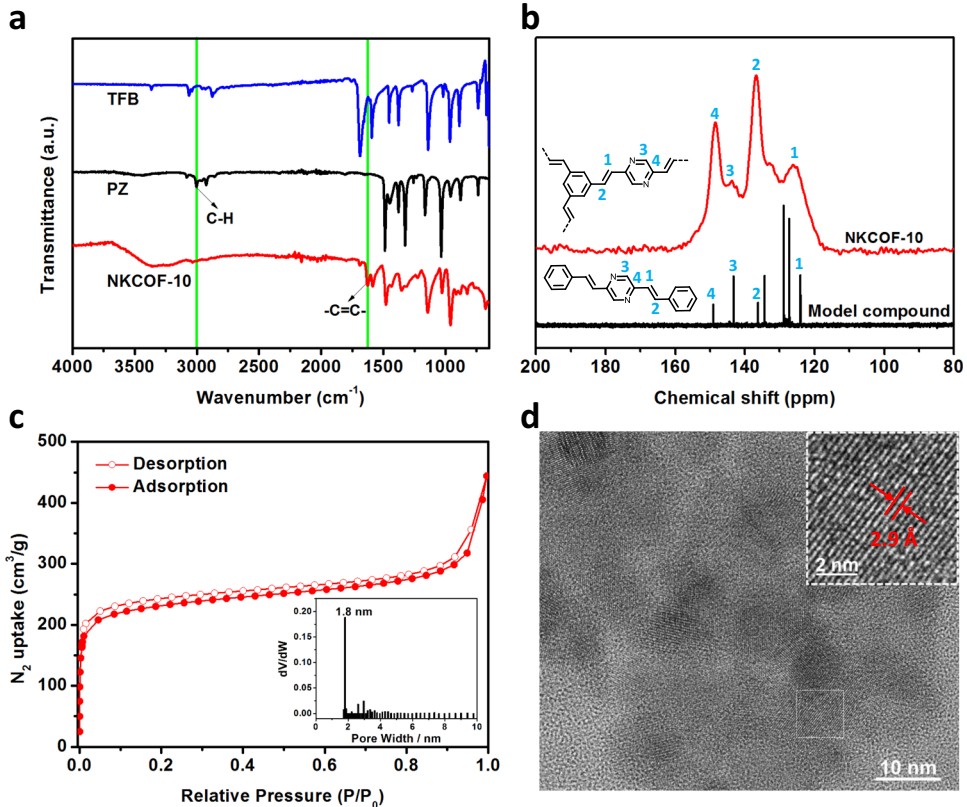

**Fig. 3 Physical characterization of NKCOF-10. a** FT-IR spectra of NKCOF-10 and corresponding monomers. **b** $^{13}$C solid-state NMR spectra of NKCOF-10 and liquid $^{13}$C NMR spectra of the model compound. **c** Nitrogen adsorption and desorption isotherms of NKCOF-10. Inset is the pore-size distribution. **d** HRTEM image of NKCOF-10.

The permanent porosity and surface area of NKCOF-10 were evaluated by N$_2$ sorption measurement at 77 K (Fig. 3c). The adsorption isotherm curse showed a sharp rise in the low relative pressure range ($P/P_0 < 0.05$), which can be described as a type I sorption isotherm, suggesting its microporous structure. The Brunauer–Emmett–Teller (BET) and Langmuir surface areas of NKCOF-10 were calculated to be 811 and 1215 m$^2$ g$^{-1}$, respectively (Supplementary Fig. 8). The pore size distribution was calculated by the nonlinear density functional theory (NLDFT) cylindrical pore model. The result displayed a prominent pore size distribution profile centered at 1.8 nm, consistent with the calculated pore size (1.9 nm) for the eclipsed AA layer stacking model. Thermogravimetric analysis (TGA) revealed the high thermal stability of NKCOF-10, which showed no significant weight loss up to 400 °C (Supplementary Fig. 9). The chemical stability of NKCOF-10 was examined via treating with boiling water, sodium hydroxide (NaOH, 10 M), and hydrochloric acid (HCl, 12 M), respectively. The crystallinity and skeletal structures can be retained, as verified by FT-IR and PXRD (Supplementary Fig. 10). The high porosity and excellent chemical stability of NKCOF-10 provide a powerful guarantee for proton conductivity applications.

Solvent-free synthesis is usually carried out under mild operation conditons (e.g., ordinary pressure) that are desirable for the large-scale production of materials. Therefore, one-pot gram-scale synthesis of NKCOF-10 can be facilely achieved under the same reaction condition (Supplementary Figs. 11 and 12). Notably, NKCOF-10 of a cylinder shape (1.2 g, height of 0.8 cm, diameter of 2.0 cm) was obtained, which showed good crystallinity and high surface area (BET: 902 m$^2$ g$^{-1}$; Langmuir: 1307 m$^2$ g$^{-1}$).

**Proton conductivity**. High water adsorption capacity is essential to proton conductivity under humidity. Water vapor isotherm (Supplementary Fig. 13) revealed that NKCOF-10 adsorbed 635 cm$^3$ g$^{-1}$ water vapor at 90% RH and 298 K. The N atoms in pyrazine groups could serve as primary adsorption sites for water molecules since hydrogen bonding (O–H···N) between pyrazine groups and water molecules has been observed in previous studies[40]. Water contact angle measurement further confirmed the high hydrophilicity for NKCOF-10 (Supplementary Fig. 14). Thus, NKCOF-10 could possess intrinsic proton conductivity after adsorbing water, which was evaluated via electrochemical impedance spectroscopy (EIS). Although negligible proton conductivity was observed for NKCOF-10 in anhydrous condition, the proton conductivity value increased dramatically under the humidity conditions and achieved $1.08 \times 10^{-5}$ S cm$^{-1}$ at 80 °C under 90% RH (Fig. 4a and Supplementary Table 2). These results indicated that water molecules played a crucial role in the proton conduction process, consistent with the literature results[41,42].

Because heteroaromatic nitrogen atoms can act as protonation sites[43], pyrazine groups in NKCOF-10 possess the potential to bind with proton carriers such as H$_3$PO$_4$. In order to further improve the proton conductivity, H$_3$PO$_4$ was successfully loaded into the channel of NKCOF-10 (termed as H$_3$PO$_4$@NKCOF-10) by the traditional immersing method. Notably, the COF powders exhibited a pronounced color change from greenish-yellow to red upon exposure to H$_3$PO$_4$ solutions (Fig. 4b), accompanied by a redshift of adsorption band in the solid-state Ultraviolet-visible spectrum (Supplementary Fig. 15)[44–46]. FT-IR spectroscopy further evidenced the protonation of the pyrazine (Supplementary Fig. 16). In the protonated structure, the typical absorption peaks of the pyrazine units shifted from 1479 to 1481 cm$^{-1}$

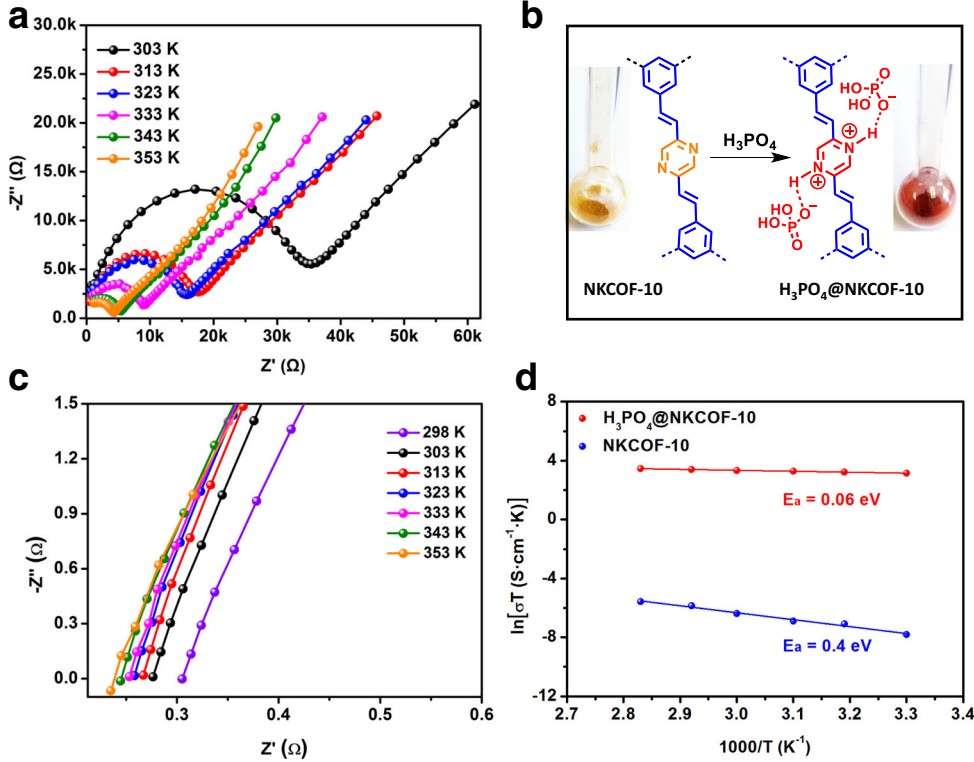

**Fig. 4 Proton transport and activation energy. a** Nyquist plots of NKCOF-10 measured under 90% RH at different temperatures. **b** Protonation of the pyrazine moieties with $H_3PO_4$. **c** Nyquist plots of $H_3PO_4$@NKCOF-10 under 90% RH at different temperatures. **d** Arrhenius plots for NKCOF-10 (blue) and $H_3PO_4$@NKCOF-10 (red).

(stretching mode), and from 1350 to 1360 $cm^{-1}$ (breathing mode). The $H_3PO_4$ loading efficiency in $H_3PO_4$@NKCOF-10 was calculated to be ~31 wt% after weighing. The scanning electron microscopy (SEM) energy-dispersive spectroscopy (EDX) mapping images demonstrated that $H_3PO_4$ was homogeneously distributed throughout the COF particles (Supplementary Fig. 17). The BET surface area of $H_3PO_4$@NKCOF-10 showed a dramatic decrease, which further confirmed the loading of $H_3PO_4$ molecules in the channel of the COF (Supplementary Fig. 18). Although the crystallinity of NKCOF-10 decreased due to the loading of amorphous guests (i.e., $H_3PO_4$) in the channels, the crystallinity can fully recover after washing with a saturated aqueous $NaHCO_3$ solution, water, and alcohol (Supplementary Fig. 19).

Next, we investigated the proton conductivity of $H_3PO_4$@NKCOF-10 from 298 to 353 K under 90% RH, and the Nyquist plots were shown in Fig. 4c. $H_3PO_4$@NKCOF-10 had a proton conductivity of $6.97 \times 10^{-2}$ S $cm^{-1}$ at 298 K, which is the highest value among all reported COFs under the same conditions (Supplementary Table 3). With the increase of temperature, the proton conductivity value continued to rise and reached up to $9.04 \times 10^{-2}$ S $cm^{-1}$ at 353 K (Supplementary Table 2), which was almost four orders of magnitude higher than the pristine NKCOF-10 without loading $H_3PO_4$. The activation energy ($E_a$) value of proton conduction for $H_3PO_4$@NKCOF-10 was 0.06 eV calculated from temperature-dependent Arrhenius plots (Fig. 4d). This value was much smaller than the pristine NKCOF-10 ($E_a$ = 0.4 eV), Nafion film (0.22 eV), and all reported COFs (Supplementary Table 3)[47], indicating the Grotthuss mechanism ($E_a$ < 0.4 eV) rather than the vehicular-type mechanism ($E_a$ > 0.4 eV)[48]. To further evident the critical role of water molecules in the proton transport mechanism, the humidity-dependent proton conductivity of $H_3PO_4$@NKCOF-10 was measured at 50 °C

(Supplementary Fig. 20). The proton conductivity value drastically increased from $3.77 \times 10^{-4}$ to $7.8 \times 10^{-2}$ S $cm^{-1}$ when RH raised from 10% to 90%. The water adsorption isotherms and hydrophobic angle measurements indicated high water uptake capacity and hydrophilicity property for $H_3PO_4$@NKCOF-10 (Supplementary Figs. 13 and 21). Thus, the adsorbed water molecules can assist the formation of hydrogen-bonding networks in the COF channels via P=O···H–O and O–H···N=C interactions in nanochannels, which supported a hydrogen-bond-mediated proton transport mechanism. In addition, the high proton conductivity of $H_3PO_4$@NKCOF-10 can be retained upon continuous run over 2 days (Supplementary Fig. 22), indicative of no leakage of $H_3PO_4$ and high conductivity stability. The content of $H_3PO_4$ was maintained after EIS measurements by TGA analysis (Supplementary Fig. 23), further confirming no leakage of $H_3PO_4$ during the EIS measurements.

The ultrahigh proton conductivity of $H_3PO_4$@NKCOF-10 can be comparable to traditional Nafion materials[48], making it a promising candidate as solid-state electrolytes in PEMFCs. Fabrication of COF membranes with good mechanical properties is essential for practical application as fuel cell membranes[49]. Thus, we mixed a small amount of PTFE (3 wt%) into $H_3PO_4$@NKCOF-10 as a binder to fabricate a large-scale, self-standing membrane (Supplementary Figs. 24 and 25)[50]. The formed membrane was used as solid-state electrolytes for the construction of PEM fuel cells under $H_2/O_2$ operation conditions (Fig. 5a). The anode and cathode layers were fabricated using a commercial 40% Pt/C catalyst, and the Pt loading was maintained as 1 mg $cm^{-1}$ on each electrode. As shown in Fig. 5b, the single-cell assembly exhibited an open circuit voltage (OCVs) of 0.87 V, indicating that the composite membrane could be in good contact with electrodes and no fuel gas leak in this system. Notably, $H_3PO_4$@NKCOF-10 shows a maximum power density of 135

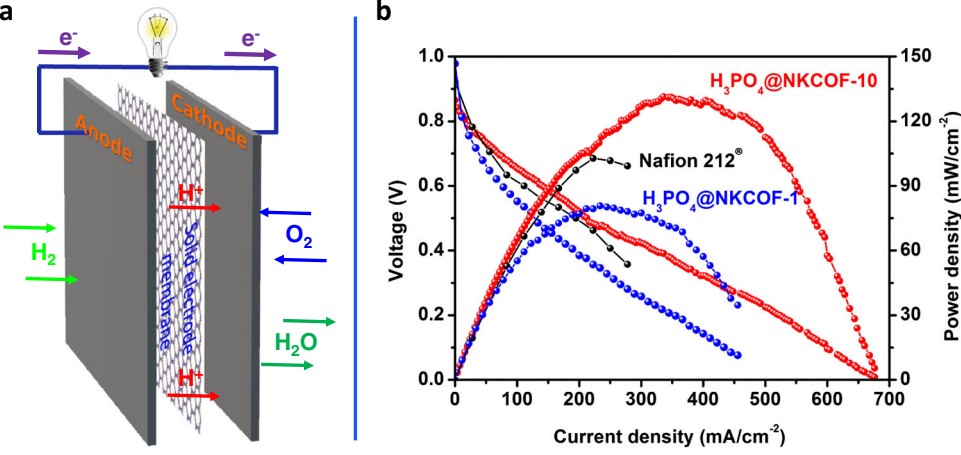

**Fig. 5 Fuel cell of H₃PO₄@NKCOF-10. a** Scheme of the PEMFCs using H₃PO₄@NKCOF-10, H₃PO₄@NKCOF-1, and Nafion 212 as solid electrolyte membranes of MEA. **b** Polarization curves and power density curves obtained at 60 °C and 1.0 atm (H₂ and O₂) under 100% RH condition.

mW cm$^{-2}$ and a maximum current density of 676 mA cm$^{-2}$ at 323 K, which are higher than the cell with Nafion membrane and the benchmark COF material (i.e., H₃PO₄@NKCOF-1)[27]. These results further suggested that H₃PO₄@NKCOF-10 was a promising, highly conductive material for PEMFC applications.

In conclusion, we developed a green synthesis strategy to synthesize a highly crystalline olefin-linked COF through a benzoic-anhydride-catalyzed Aldol reaction under solvent-free conditions. The formed COF (NKCOF-10) possessed a honeycomb-like layered framework with high surface areas and exhibited exceptional stability toward harsh conditions (e.g., strong acid and base). The nitrogen sites on the pore wall were exploited to confine and stabilize the H₃PO₄ within the 1D nanochannels that afforded both ultrahigh proton conductivity ($9.04 \times 10^{-2}$ S cm$^{-1}$) and ultralow activation energy (0.06 eV). Noteworthily, the proton conductivity ($6.97 \times 10^{-2}$ S cm$^{-1}$) at 298 K for H₃PO₄@NKCOF-10 was a new record for all reported COFs under the same conditions. Furthermore, a realistic proton exchange membrane fuel cell using H₃PO₄@NKCOF-10 as the solid electrolyte reached values up to 135 mW cm$^{-2}$ and 676 mA cm$^{-2}$ for maximum power and current density, respectively, the best performance for COF materials reported so far. Further studies to explore the generality of this green synthesis approach and apply the olefin-linked COFs for photoactuator and membrane separation application are ongoing in our group. We anticipate that this study will open up new possibilities for the green synthesis of advanced materials, especially COFs, and provide important guidance for the rational design and synthesis of polymeric materials for proton-exchange membrane fuel cell applications.

## Methods

**General**. Unless otherwise stated, all materials were commercially available and used without further purification. All solvents were of analytical grade and used without further purification. PZ and TFB were purchased from HEOWNS. Benzoic anhydride was purchased from Alfa Aesar.

**Characterization**. Solid-state NMR experiments were performed on Varian Infinityplus 300 solid-state NMR spectrometer (300 MHz) using a 4 mm double resonance MAS probe. ¹³C NMR spectra were collected using the rotor frequency of 10 kHz with a contact time of 2 ms (ramp 100) and a pulse delay of 3 s. The cross polarization time was 1 ms ¹H NMR spectra was recorded on Bruker AV400 instruments at 400 MHz. Chemical shifts were reported in parts per million (ppm) downfield from internal tetramethylsilane. The surface areas of tested materials were determined using a Micromeritics ASAP-2046. Pore size distributions and pore volumes were derived from the adsorption branches of the isotherms using the NLDFT pore model for pillared clay with cylindrical pore geometry. PXRD patterns of all the materials were collected at ambient temperature on Rigaku $d_{max}$

2500 diffractometer using Cu Kα ($\lambda$ = 1.5418 Å) radiation, with a scan speed of 1°/min, a step size of 0.02° in 2$\theta$, and a 2$\theta$ range of 2–40°. The absorption of water vapor was screened using a Micromeritics ASAP-2020. FT-IR spectra were recorded on a Nicolet iS 50 ATR-FTIR instrument. Ultraviolet–visible absorption spectra of solution samples are collected using an Agilent Cary 100 UV/Vis spectrophotometer with background correction. SEM images were taken with Hitachi JSM-7500F SEM. HRTEM images were characterized on a FEI Talos F200X G2 electron microscope. The electrochemical workstation, CompactStat, IVIUM Tech., was applied to measure the proton conductivity. Hydrophobic angle measurement was recorded using a SINDIN SDC-200.

**Synthesis of NKCOF-10**. In a typical synthesis, TFB (16.2 mg, 0.1 mmol), PZ (16.2 μL, 0.15 mmol), and benzoic anhydride (45 mg, 0.2 mmol) were weighed into a Pyrex tube. The tube was degassed by the freeze–pump–thaw technique three times and sealed under vacuum. Then the tube was transferred into an oven to heat at 200 °C for 5 days yielding a greenish brown solid. The greenish brown solid was collected then solvent exchanged with THF and methanol, and dried at 100 °C under vacuum for 12 h to afford NKCOF-10.

**Bulky production of NKCOF-10**. TFB (648 mg, 4.0 mmol), PZ (649 μL, 6.0 mmol), and benzoic anhydride (2714 mg, 12.0 mmol) were added into a 25 mL Pyrex tube. The tube was degassed by the freeze–pump–thaw technique three times and sealed under vacuum. Then the tube was treated at 200 °C for 5 days to finish the solvent-free condensation. The cylinder shape material was collected and washed with THF and methanol, and dried at 100 °C under vacuum for 12 h. The weight of purified NKCOF-10 was 1.2 g.

**Synthesis of H₃PO₄@NKCOF-10**. Dried NKCOF-10 powders sample ($W_1$, mg) were soaked in 4 M H₃PO₄ for 24 h at room temperature. Centrifugation and removal of the solvent, and then dried at 120 °C for 10 h to obtain dry H₃PO₄@NKCOF-10, the weight ($W_2$, mg). The amount of H₃PO₄ incorporated into COF was calculated by the following equation:

$$\text{H}_3\text{PO}_4 \text{ uptake}(\%) = \frac{W_2 - W_1}{W_2} \times 100\% \qquad (1)$$

The measurement was repeated three times and the standard deviation was within ±2.0%.

**Proton conductivity measurements**. Proton conductivity of the COFs was measured by AC impedance using Ivium CompactStat potentiostat B31250 under controlled humidity and temperature. About 50 mg materials were pressed into circular pellets (13 mm in diameter, 200–400 μm in thickness) under a pressure of 30 MPa for 15 s. The rectangular pellets were placed between two-electrode cell connected with Ivium CompactStat potentiostat B31250 by a conductive wire. The temperature dependence of proton conductivity was tested by EIS with a tuned frequency range from 1 Hz to 1 MHz and an alternating potential of 100 mV in a humidity chamber maintained at 90% RH. The humidity dependence of proton conductivity was determined using different humidities controlled by saturated salt aqueous solutions in a constant temperature and humidity chamber. When changing temperature or humidity, the pellets were equilibrated for 2 h. Proton conductivity ($\sigma$, S cm$^{-1}$) was calculated by the following equation:

$$\sigma = L/(RA) \qquad (2)$$

where $\sigma$ is the proton conductivity (S cm$^{-1}$), $L$ is the thickness of the pellet (cm),

$A$ is the area of the pellet (cm$^2$), and $R$ is the resistance $(\Omega)$ of the pellet corresponding to the real $Z'$ Nyquist plot. An equivalent circuit (shown below) is adopted to fit the data in $Z$-plot software.

**Proton exchange membrane fuel cells**. Fabrication of the anode and the cathode electrodes: Weighing 30 mg of Pt/C (containing 40% of Pt) and dispersing it into a mixture solvent of isopropanol (1 mL), water (150 µL), and Nafion (500 µL, 5 wt% in water) to form a uniform suspension. Then, coating the suspension onto a 12 cm$^2$ (3 cm × 4 cm) carbon paper, dried in the oven. So the Pt-loading controlled was controlled as 1.0 mg cm$^{-2}$. The MEA was prepared by sandwiching the H$_3$PO$_4$@NKCOF-10 membrane between the anode and cathode electrodes with maintaining an active area of 2 cm × 2 cm for the cathode electrodes (for O$_2$ passage). Because the area of H$_3$PO$_4$@NKCOF-10 membrane was smaller than that of the air-flow passages of the test device, we used an active area of 0.6 cm × 0.6 cm for the anode electrodes (for H$_2$ passage), with other regions kept airtight. The measurement was carried out using standard PEMFCs protocol in a 100% humid H$_2$ and O$_2$ environment where the gas flow was 50 mL/min to both the anode and cathode. The test was carried out without any back pressure. The measurement was operated at a cell temperature of 60 °C.

## Data availability

All data supporting the findings of this study are available within the article, as well as the Supplementary Information file, or available from the corresponding authors on reasonable request.

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

## Acknowledgements

The authors acknowledge the National Key Research and Development Program of China (2020YFA0907300), 111 projects (B12015), and Postdoctoral Science Foundation of China (2019M660974). We also thank Professor Zhongyi Jiang's team from Tianjin University for the assistance in evaluating the PEMFC performance. Dedicated to the 100th anniversary of Chemistry at Nankai University.

## Author contributions

Z.Z. conceived and designed the project. Z.W., Y.Y., and Zf.Z. performed the experiments. P.Z., Y.Z., and J.L. helped with the structural characterization analysis. Y.C., and P.C. helped to analyze the results of proton conductivity tests. S.M. helped to simulate the structure of COFs. Z.Z. and Z.W. wrote the manuscript with contributions from all the authors.

## Competing interests

The authors declare no competing interests.
