## [Peer Review File · Nature Communications]

REVIEWER COMMENTS

Reviewer #1 (Remarks to the Author):

The paper describes the synthesis of olefin-linked COF by use of unique solvent-free protocol, and studied proton conductivity by immersing H₃PO₄ inside the channels. They studied high proton conductivity with structural integrity and applied the material for PEMFC device to have sufficient I-V performance (with PTFE).

Recently there are many papers on high proton conductivity with COFs by using acid doping such as H₂SO₄, H₃PO₄. The approach has been studied from various past works, and the current COF has one of highest chemical stability and resulting highest proton conductivity which is the benefit of olefin-linking. They characterized structures and mechanism of proton conductivity by many ways and the discussion is mostly reliable. One novel point is that they further studied the real PEMFC device which has not been done by other COF reports. Although they used PTFE as a filler, the device seems to work well under the hydrated condition. I have several questions and requests for revision as shown in below, but in principle, the work explores new synthesis of highly-performing COF having acidic sites with H₃PO₄, and high standard characterization which are good impact for materials chemistry and electrochemistry. There are several misleading discussion and points to be further characterized, but I think this is worth to be published in this journal.

- Even though they titled "green synthesis" of this COF because of solvent-free procedure, the synthesis needs 200 degC for 5 days which is highly energy-consumption process. It is no need to highlight "green synthesis" here. I wonder the possibility on the synthesis of this COF by mechanochemical route, have you tried mechanochemistry?
- Decrease the peak intensity upon H₃PO₄ accommodation is interesting behavior. Can they check how the overall structure loses long-range crystallinity? This is related to the elastic behavior of the COF.
- How is the H⁺ conductivity for H₃PO₄@NKCOF-10 under the anhydrous condition?
- Low activation energy for H₃PO₄@NKCOF-10 is surprisingly low. On the other hand, we also found the activation energy (0.4 eV) for non-doped NKCOF-10 is also low in the observed conductivity regime in Figure 3d. Can they explain the reason of the low activation energy as well?
- In the section of PEMFC performance: they say "indicating that the composite membrane had high proton conductivity and mechanical properties " from OCV=0.87 V, but this is not an appropriate discussion. It is better to say no fuel gas leak and good contact with electrodes in this system to observed stable OCV.
- Comparison of the performances in Figure 4c is not fair because each previous report demonstrated different condition and the graph only shows COF-related compounds. Conventional organic polymer-based FC shows obviously much higher values, and the current graph would mislead the state-of-art which does not match the scope of Nat Comm. I recommend deleting this graph. (actually Table 2 is not fair, too)

- Experimental details are missing for SSNMR (measurement condition and sequence used), fabrication of fuel cell (film thickness, materials of cathode/anode, purities of H₂/O₂, humidity). Please add the information to make clear the conditions tried.

- TGA profile in Figure S7 requires the information of measurement atmosphere (N₂?)

- Figure S11 is not necessary.

- Scale bars in Figure S15 are too small to read.

- Scheme 1a typo : Perious

Reviewer #2 (Remarks to the Author):

Zhang and co-workers report the synthesis of new C=C linked COF and its application as a proton-conductive membrane in H₂/O₂ fuel cells. The key elements of novelty are as follow:

1) Using benzoic anhydride as a promotor enables aldol polycondensation of a relatively inert dimethylpyrazine, which expands the scope of C=C linked COFs (Scheme 1) – and thus a greater variety of properties. It also allows for solvent-free synthesis with benzoic acid as the only by-product, which is of potential interest for large-scale synthesis.

2) When doped with phosphoric acid, the resulting COF displays high proton conductivity (0.07 S/cm at RT). The very low activation energy (0.06 eV) of conduction is notable as it allows for efficient operation at RT. The H₂/O₂ (Pd/C) fuel cell constructed with the reported COF shows maximum power of 0.14 W/cm² which appears to be the highest value reported for fuel cells with COF-based membranes.

I believe the synthesis and the device application present substantial advances in the field and the paper may become suitable for Nat. Commun. after a revision on the points below:

1) In the context of green synthesis, the authors should mention the previous work of Banerjee on mechanochemical synthesis of COFs via screw-extrusion. Also, while the introduction provides a fair overview of the related work on C=C linked COFs, it omits to mention that high proton conductivity in a similar triazine-based COF has been reported in ref. 35.

2) The reported synthetic procedure is for an extremely small scale reaction (<30mg of COF), even though ~50 mg samples are used in many of the reported analyses. Both precursors are readily available and not too expensive, and I strongly recommend the authors to repeat the synthesis on a substantial (at least 1 g) scale to show that the yield of the COF and the crystallinity/purity (PXRD/FTIR) are not detrimentally affected. Otherwise, the notion of “green synthesis” seems irrelevant.

3) The text refers to (110) and (001) peaks at 7.65 and 25.7 θ , respectively. I don't see these in the experimental PXRD (Fig.1b). If observable, show an expansion in SI. Also, pls report FWHM for (100) peak.

4) The residual FTIR peak of the unreacted aldehyde group is appreciable (ca. ~10% of relative intensity vs the starting trialdehyde). This translates to one-two CHO end-groups per each hexagonal

pore of the COF, and the statement of “high polymerization degree” is not justified. This could be due to inexact stoichiometry while working at the low-mg scale; might improve on a large scale.

5) Fig. 2d: give the unit for the scale-bar in the inset. What crystallographic planes the 2.7Å spacing corresponds to? Do the other striped features in the same TEM correspond to the same 2.7Å spacing?

6) I doubt TGA is well-suited for the quantification of H₃PO₄ loading. First, the phosphate could become covalently linked to the COF upon heating >300°C. Second, the presence of H₃PO₄ may accelerate the degradation of COF <400°C. Why don't the authors use a simple gravimetry (re-weighting the COF after immersing in H₃PO₄ and drying)?

7) Suggested di-protonation of pyrazine (Fig. 3b) is unrealistic; only monoprotonation could be expected based on the pK_a's, and <<1 H₃PO₄ per pyrazine ring is implied by the 16% loading ratio.

8) The claim of the “new record” of power for COF-based fuel cells is significant but incomplete. The details of the measurements (eg, what gas pressure was used? How was Pt loading controlled and measured – 1 mg cm⁻¹(??) from “40% Pt/C catalyst”?) are not provided but critically important. Discussion of the highest previously reported PEMFC figures of merit should be given (including the author's own ref. 27). A comparison with the commercial PEM (eg, Nafion) under the same conditions, is needed.

9) Some minor points: where is the data for the reported Langmuir surface area (the cited Fig.S6 only contains BET); too many significant figures for the conductivity values (implies unrealistic accuracy); corrupted symbols in the text (probably, for °C); in the synthesis of NKCOF-10, does 'TB' refer to dimethylpyrazine?? (shown as PZ elsewhere); the UV-Vis data is not of high quality (probably over-saturated, consider diluting the sample with KBr); the English usage requires some attention.

Reviewer #3 (Remarks to the Author):

The application of covalent organic framework materials in proton conduction and fuel cell has been a hot topic. Wang et al. prepared a covalent organic framework of olefin linkage by green synthesis, and tested and simulated the structure by various means. Phosphoric acid molecules are further anchored within the framework. The comparison showed that the proton conductivity of the modified compound was significantly improved. Finally, the performance of the simulated fuel cell is tested. Generally speaking, it meets the requirements of publication. It is suggested that it be published with the following minor modifications.

(1) please carefully check the grammar and format. For example, in Scheme 1, perious should be previous; In many places in the MS, the temperature symbol is lost (lines, 72,131,137,180,203,248 at so on); please check the format of references.

(2) The green synthesis of covalent organic frameworks has been reported before. Please give a brief overview in the Introduction.

Reviewer #1:

Comments 1: The paper describes the synthesis of olefin-linked COF by use of unique solvent-free protocol, and studied proton conductivity by immersing H₃PO₄ inside the channels. They studied high proton conductivity with structural integrity and applied the material for PEMFC device to have sufficient I-V performance (with PTFE).

Recently there are many papers on high proton conductivity with COFs by using acid doping such as H₂SO₄, H₃PO₄. The approach has been studied from various past works, and the current COF has one of highest chemical stability and resulting highest proton conductivity which is the benefit of olefin-linking. They characterized structures and mechanism of proton conductivity by many ways and the discussion is mostly reliable. One novel point is that they further studied the real PEMFC device which has not been done by other COF reports. Although they used PTFE as a filler, the device seems to work well under the hydrated condition. I have several questions and requests for revision as shown in below, but in principle, the work explores new synthesis of highly-performing COF having acidic sites with H₃PO₄, and high standard characterization which are good impact for materials chemistry and electrochemistry. There are several misleading discussion and points to be further characterized, but I think this is worth to be published in this journal.

Response: We thank the reviewer for the comments and support of our work.

Comments 2: Even though they titled "green synthesis" of this COF because of solvent-free procedure, the synthesis needs 200°C for 5 days which is highly energy-consumption process. It is no need to highlight "green synthesis" here. I wonder the possibility on the synthesis of this COF by mechanochemical route, have you tried mechanochemistry?

Response: We thank the reviewer for this suggestion. "Green chemistry" has been clearly defined by the IUPAC and can be summarized as "The invention, design, and application of chemical products and processes to reduce or to eliminate the use and generation of hazardous substances." In our manuscript, solvent-free synthesis can effectively avoid the use of hazardous organic solvents (e.g., DMF, Methanol, Mesitylene), which meet the requirement of green synthesis. In addition, solvent-free synthesis conditions can avoid high-pressure operation during the reaction, which is desirable for the large-scale production of high-quality COFs and, in principle, scalable to any amount depending on the requirement. To demonstrated the advantage of this green synthesis route, we have successfully achieved a one-pot gram-scale synthesis of high-quality COF (i.e., NKCOF-10, data shown in Supplementary Figure 10 and 11). The large-scale production of high-quality COFs is crucial to reduce energy consumption that meets the criteria of green synthesis.

Moreover, according to the reviewer's suggestions, we have tried the mechanochemical route, but failed. It is possibly because Aldol condensation reactions shown in this study are usually conducted under harsh conditions (J. Am. Chem. Soc. 2019, 141, 6848; J. Am. Chem. Soc. 2020, 142, 8862; Angew. Chem. Int. Ed. 2019, 58, 13753).

Comments 3: Decrease the peak intensity upon H₃PO₄ accommodation is interesting behavior. Can they check how the overall structure loses long-range crystallinity? This is related to the elastic behavior of the COF.

Response: We thank the reviewer for this suggestion. The Introduction of H₃PO₄ caused a decrease of the crystallinity (in the PXRD pattern, the (100) plane intensity decreased, the (001) plane intensity increased), mainly because the pores of COF are occupied by disordered H₃PO₄ with strong interaction with host structures. The same phenomenon is observed in the literature (e.g., J. Am. Chem. Soc. 2020, 142, 14357–14364; J. Mater. Chem. A, 2016, 4, 2682-2690). Not surprisingly, the

crystallinity can be recovered via washing the H₃PO₄ molecules out (in the PXRD pattern, the (100) plane intensity increased, the (001) plane intensity decreased) (Supplementary Figure 18).

Comments 4: How is the H⁺ conductivity for H₃PO₄@NKCOF-10 under the anhydrous condition?

Response: We thank the reviewer for this suggestion. The anhydrous proton conductivity of H₃PO₄@NKCOF-10 is 4.1×10^{-5} S cm⁻¹ at 413 K (Supplementary Figure 25), which is lower than H₃PO₄@NKCOF-10 tested under humidity conditions.

Comments 5: Low activation energy for H₃PO₄@NKCOF-10 is surprisingly low. On the other hand, we also found the activation energy (0.4 eV) for non-doped NKCOF-10 is also low in the observed conductivity regime in Figure 3d. Can they explain the reason of the low activation energy as well?

Response: We thank the reviewer for this suggestion. As is reported in the literature (J. Am. Chem. Soc. 2017, 139, 10079-10086; Coord. Chem. Rev. 2020, 422, 213465), water molecules play a crucial role in the proton conduction process, and the existence of hydrogen-bonded networks will accelerate proton transport. NKCOF-10 possess high water vapor adsorption capacity (635 cm³ g⁻¹) and good hydrophilicity, identified by water vapor adsorption at 298 K (Supplementary Figure 12) and hydrophobic angle measurements (Supplementary Figure 13), which can facilitate the proton conduction. In addition, NKCOF-10 possesses well-organized one-dimensional nanochannels with abundant nitrogen sites. The regular channels can shorten the transport distance, and abundant nitrogen sites can facilitate the formation of hydrogen-bonded networks with water molecules in the confined channels. All these features can attribute to the low conduction energy barrier of proton, i.e., low activation energy.

Comments 6: In the section of PEMFC performance: they say "indicating that the composite membrane had high proton conductivity and mechanical properties " from OCV=0.87 V, but this is not an appropriate discussion. It is better to say no fuel gas leak and good contact with electrodes in this system to observed stable OCV.

Response: We thank the reviewer for this suggestion. Per the suggestion, we have corrected the description in the main text.

Comments 7: Comparison of the performances in Figure 4c is not fair because each previous report demonstrated different condition and the graph only shows COF-related compounds. Conventional organic polymer-based FC shows obviously much higher values, and the current graph would mislead the state-of-art which does not match the scope of Nat Comm. I recommend deleting this graph. (actually Table 2 is not fair, too)

Response: We thank the reviewer for this suggestion. Per the suggestion, we have deleted them.

Comments 8: Experimental details are missing for SSNMR (measurement condition and sequence used), fabrication of fuel cell (film thickness, materials of cathode/anode, purities of H₂/O₂, humidity). Please add the information to make clear the conditions tried.

Response: We thank the reviewer for this suggestion. According to the reviewer's suggestions, we added the related description in the revised main text.

Comments 9: TGA profile in Figure S7 requires the information of measurement atmosphere (N₂?)

Response: We thank the reviewer for this suggestion. According to the reviewer's suggestions, we have added the information of measurement atmosphere.

Comments 10: Figure S11 is not necessary.

Response: We thank the reviewer for this suggestion. According to the reviewer's suggestions, we have deleted Figure S11.

Comments 11: Scale bars in Figure S15 are too small to read.

Response: We thank the reviewer for this suggestion. This graph has been enlarged according to the reviewer's suggestions.

Comments 12: Scheme 1a typo: Perious

Response: We thank the reviewer for the suggestion. According to the reviewer's suggestions, We have corrected the typo.

Reviewer #2:

Comments 1: Zhang and co-workers report the synthesis of new C=C linked COF and its application as a proton-conductive membrane in H₂/O₂ fuel cells. The key elements of novelty are as follow:

1) Using benzoic anhydride as a promotor enables aldol polycondensation of a relatively inert dimethylpyrazine, which expands the scope of C=C linked COFs (Scheme 1) – and thus a greater variety of properties. It also allows for solvent-free synthesis with benzoic acid as the only by-product, which is of potential interest for large-scale synthesis.

2) When doped with phosphoric acid, the resulting COF displays high proton conductivity (0.07 S/cm at RT). The very low activation energy (0.06 eV) of conduction is notable as it allows for efficient operation at RT. The H₂/O₂ (Pd/C) fuel cell constructed with the reported COF shows maximum power of 0.14 W/cm² which appears to be the highest value reported for fuel cells with COF-based membranes.

I believe the synthesis and the device application present substantial advances in the field and the paper may become suitable for Nat. Commun. after a revision on the points below:

Response: We thank the reviewer for the comments and support of our work.

Comments 2: In the context of green synthesis, the authors should mention the previous work of Banerjee on mechanochemical synthesis of COFs via screw-extrusion. Also, while the Introduction provides a fair overview of the related work on C=C linked COFs, it omits to mention that high proton conductivity in a similar triazine-based COF has been reported in ref. 35.

Response: We thank the reviewer for the suggestions. Per the suggestions, we added more related descriptions in the revised main text.

Comments 3: The reported synthetic procedure is for an extremely small scale reaction (<30 mg of COF), even though ~50 mg samples are used in many of the reported analyses. Both precursors are readily available and not too expensive, and I strongly recommend the authors to repeat the synthesis on a substantial (at least 1 g) scale to show that the yield of the COF and the crystallinity/purity (PXRD/FTIR) are not detrimentally affected. Otherwise, the notion of "green synthesis" seems irrelevant.

Response: We thank the reviewer for the suggestions. Per the suggestions, we have successfully obtained gram-scale NKCOF-10 (1.2 g) in one pot under the same condition. Various characterizations, including PXRD pattern, FT-IR spectroscopy, and nitrogen adsorption and desorption isotherms (Supplementary Figure 10 and 11) revealed the gram-scale NKCOF-10 showed good crystallinity and high surface area comparable to small-scale synthesis. These results demonstrated the advantage of this green synthesis route. The related description and data have been added to the main text and SI.

Comments 4: The text refers to (110) and (001) peaks at 7.65 and 25.7°, respectively. I don't see these in the experimental PXRD (Fig.1b). If observable, show an expansion in SI. Also, pls report FWHM for (100) peak.

Response: We thank the reviewer's kind comments and suggestions. We have shown an expansion of PXRD and FWHM in the revised Supplementary Information (Supplementary Figure 6).

Comments 5: The residual FTIR peak of the unreacted aldehyde group is appreciable (ca. ~10% of relative intensity vs the starting trialdehyde). This translates to one-two CHO end-groups per each hexagonal pore of the COF, and the statement of "high polymerization degree" is not justified. This could be due to inexact stoichiometry while working at the low-mg scale; might improve on a large scale.

Response: We thank the reviewer's suggestions. Per the suggestion, we have revised the statement to be 'indicative of polymerization'. Moreover, we also tried a large scale synthesis. Based on the FTIR data, the relative intensity of the absorption peak of the unreacted aldehyde group slightly decreased, indicative of higher polymerization degree, but still not fully react.

Comments 6: Fig. 2d: give the unit for the scale-bar in the inset. What crystallographic planes the 2.7 Å spacing corresponds to? Do the other striped features in the same TEM correspond to the same 2.7 Å spacing?

Response: We thank the reviewer for this suggestion. According to the suggestion of the reviewer, we referred to the literature (Angew. Chem. Int. Ed. 2019, 58, 13753 -13757; J. Am. Chem. Soc. 2019, 141, 5880–5885.) and remeasured the interlayer distance to be ~3.4 Å, corresponding to the (001) facet, which was in good accordance with that obtained from PXRD refinement.

Comments 7: I doubt TGA is well-suited for the quantification of H₃PO₄ loading. First, the phosphate could become covalently linked to the COF upon heating >300°C. Second, the presence of H₃PO₄ may accelerate the degradation of COF <400°C. Why don't the authors use a simple gravimetry (re-weighting the COF after immersing in H₃PO₄ and drying)?

Response: Thank you for the reviewer's suggestion. According to the reviewer's suggestion, we have corrected the H₃PO₄ loading to be 31% via weighting.

Comments 8: Suggested di-protonation of pyrazine (Fig. 3b) is unrealistic; only monoprotonation could be expected based on the pK_a's, and <<1 H₃PO₄ per pyrazine ring is implied by the 16% loading ratio.

Response: We thank the reviewer's kind comments and suggestions. According to the crystal structure data reported in Acta Cryst. 2008, E64, o1289, pyrazine can be di-protonated. As mentioned in the above response (comment 7), the H₃PO₄ loading was recalculated to be 31% via weighting. Thus, both mono-protonated and di-protonated pyrazine should be present in this system.

Comments 9: The claim of the "new record" of power for COF-based fuel cells is significant but incomplete. The details of the measurements (eg, what gas pressure was used? How was Pt loading controlled and measured – 1mg cm⁻¹(??) from "40% Pt/C catalyst"?) are not provided but critically important.

Discussion of the highest previously reported PEMFC figures of merit should be given (including the author's own ref. 27). A comparison with the commercial PEM (eg, Nafion) under the same conditions, is needed.

Response: We thank the reviewer's kind comments and suggestions. In the revised main text, Methods section, we have added a paragraph on proton exchange membrane fuel cells in which the details of the measurements and Pt loading method were described. Per the suggestion, we have

added discussion and comparison with the commercial Nafion and H₃PO₄@NKCOF-1 in the revised main text.

Comments 10: Some minor points: where is the data for the reported Langmuir surface area (the cited Fig. S6 only contains BET); too many significant figures for the conductivity values (implies unrealistic accuracy); corrupted symbols in the text (probably, for °C); in the synthesis of NKCOF-10, does 'TB' refer to dimethylpyrazine?? (shown as PZ elsewhere); the UV-Vis data is not of high quality (probably over-saturated, consider diluting the sample with KBr); the English usage requires some attention.

Response: We thank the reviewer's kind comments and suggestions. According to the reviewer's suggestion, we have corrected the typos and added the required data in the revised manuscript (Supplementary Figure 7b, 14 and Table 2).

Reviewer #3:

Comments 1: The application of covalent organic framework materials in proton conduction and fuel cell has been a hot topic. Wang et al. prepared a covalent organic framework of olefin linkage by green synthesis, and tested and simulated the structure by various means. Phosphoric acid molecules are further anchored within the framework. The comparison showed that the proton conductivity of the modified compound was significantly improved. Finally, the performance of the simulated fuel cell is tested. Generally speaking, it meets the requirements of publication. It is suggested that it be published with the following minor modifications.

Response: We thank the reviewer for the comments and support of our work.

Comments 2: Please carefully check the grammar and format. For example, in Scheme 1, perious should be previous; In many places in the MS, the temperature symbol is lost (lines, 72,131,137,180,203,248 at so on); please check the format of references.

Response: We thank the reviewer's kind comments and suggestions. According to the reviewer's suggestions, we have carefully checked our manuscript and corrected the typos.

Comments 3: The green synthesis of covalent organic frameworks has been reported before. Please give a brief overview in the Introduction.

Response: We thank the reviewer's kind comments and suggestions. According to the reviewer's suggestions, we added the description of green synthesis of COFs in the revised main text.

Again we thank the reviewers for the constructive comments and suggestions, which have made our manuscript much improved.

Thank you very much for your favorable consideration of our manuscript.

Sincerely,

Zhenjie Zhang, Professor of Chemistry

REVIEWERS' COMMENTS

Reviewer #1 (Remarks to the Author):

With additional experiments, the manuscript has been improved. Some more comments for construction of manuscript are in below.

For comment 2 of reviewer 1: Definition of IUPAC could be cited.

It is better to show "This work" in above of "previous works" in Scheme 1.

Discussion and Figure 4 shows comparison of I-V profiles of COFs and Nafion. Nafion have several commercial models, it is better to clarify the model used.

Caption of Figure 4 should have "100% of O₂ and H₂ were used" of which is important to evaluate the I-V performance (not by the dried air).

Reviewer #2 (Remarks to the Author):

I am generally satisfied with the revision and the answers provided by the author. There are a few minor remaining issues, but overall the paper can be accepted for publication. Both the new COF synthesis and the fuel cell device performance would be of great interest for the field.

1) The TEM is still problematic. The authors say the new Figure 2 shows fringes with 3.4Å periodicity. However, my measurement on the provided close-up inset (in Fig. 2d) shows the spacing between the fringes around 2.9Å. Also, a ~50% larger periodicity fringes are seen in the middle left part of the TEM. While this is not a critical issue, the experimental data should be documented carefully. Please, check the scale-bar and, ideally, provide the FFT analysis in the SI to support the observed periodicity(ies).

2) The meaning of the new sentence on In. 202 seems misleading: "scale up the synthesis reaction of NKCOF-1 to reduce energy consumption". As already implied by referee 1, 200°C/5 days reaction is not exactly the best example of "low energy consumption" process. It's not the 'energy saving' but avoidance of solvent and an ambient pressure that are attractive features of the reaction.

3) Note that the symbols for degree Celsius still do not show properly (in pdf). Also, the English is still quite rough throughout the paper, sometimes to the detriment of clarity, eg. "in the absence of humidity" – means what (RH= 0%? How was it achieved?); "narrow fwhm₁₀₀ of 0.81 Å in the PXRD pattern" (you mean 0.81 degree, not angstrom, right?).

A MUCH more careful proof-read is required before publication (and before submission, really) in a prestigious journal like Nat. Comm.

Reviewer #3 (Remarks to the Author):

In the revised manuscript, the authors addressed and reflected what the reviewers pointed out. Without further correction, therefore, I recommend publishing this paper.

Reviewer #1:

Comments 1: With additional experiments, the manuscript has been improved. Some more comments for construction of manuscript are in below.

Response: We thank the reviewer for the comments and support of our work.

Comments 2: For comment 2 of reviewer 1: Definition of IUPAC could be cited.

Response: We thank the reviewer's kind comments and suggestions. According to the reviewer's suggestions, the definition of green synthesis by the IUPAC has been cited in the revised main text.

Comments 3: It is better to show "This work" in above of "previous works" in Scheme 1.

Response: We thank the reviewer's kind comments and suggestions. According to the reviewer's suggestions, we have adjusted the Scheme 1.

Comments 4: Discussion and Figure 4 shows comparison of I-V profiles of COFs and Nafion. Nafion have several commercial models, it is better to clarify the model used.

Response: We thank the reviewer's kind comments and suggestions. According to the reviewer's suggestions, we added the commercial model of Nafion 212.

Comments 5: Caption of Figure 4 should have "100% of O₂ and H₂ were used" of which is important to evaluate the I-V performance (not by the dried air).

Response: We thank the reviewer's kind comments and suggestions. According to the reviewer's suggestions, we have corrected the caption of Figure 4.

Reviewer #2:

Comments 1: I am generally satisfied with the revision and the answers provided by the author. There are a few minor remaining issues, but overall the paper can be accepted for publication. Both the new COF synthesis and the fuel cell device performance would be of great interest for the field.

Response: We thank the reviewer for the comments and support of our work.

Comments 2: The TEM is still problematic. The authors say the new Figure 2 shows fringes with 3.4 Å periodicity. However, my measurement on the provided close-up inset (in Fig. 2d) shows the spacing between the fringes around 2.9Å. Also, a ~50% larger periodicity fringes are seen in the middle left part of the TEM. While this is not a critical issue, the experimental data should be documented carefully. Please, check the scale-bar and, ideally, provide the FFT analysis in the SI to support the observed periodicity(ies).

Response: We thank the reviewer's kind comments and suggestions. According to the reviewer's suggestions, we have provided the FFT analysis and the test detail of interlayer distance in Supplementary Figure 7.

Comments 3: The meaning of the new sentence on In. 202 seems misleading: "scale up the synthesis reaction of NKCOF-1 to reduce energy consumption". As already implied by referee 1, 200°C/5 days reaction is not exactly the best example of "low energy consumption" process. It's not the 'energy saving' but avoidance of solvent and an ambient pressure that are attractive features of the reaction.

Response: We thank the reviewer's kind comments and suggestions. According to the reviewer's suggestions, we have made corresponding corrects in the revised main text.

Comments 4: Note that the symbols for degree Celsius still do not show properly (in pdf). Also, the English is still quite rough throughout the paper, sometimes to the detriment of clarity, eg. “in the absence of humidity” – means what (RH= 0%? How was it achieved?); “narrow fwhm100 of 0.81 Å in the PXRD pattern” (you mean 0.81 degree, not angstrom, right?).

Response: We thank the reviewer's kind comments and suggestions. According to the reviewer's suggestions, we have corrected it in the revised main text. “in the absence of humidity” was changed to “in anhydrous condition”, which can be achieved under N₂ atmosphere. “081 Å” was changed to “0.81°”.

Comments 5: A MUCH more careful proof-read is required before publication (and before submission, really) in a prestigious journal like Nat. Com

Response: We thank the reviewer's kind comments and suggestions. According to the reviewer's suggestions, we have carefully checked our manuscript.

Reviewer #3:

In the revised manuscript, the authors addressed and reflected what the reviewers pointed out. Without further correction, therefore, I recommend publishing this paper.

Response: We thank the reviewer for the support of our work.

Again we thank the reviewers for the constructive comments and suggestions, which have made our manuscript much improved.

Thank you very much for your favorable consideration of our manuscript.

Sincerely,

Zhenjie Zhang, Professor of Chemistry